# Axial variation of deoxyhemoglobin density as a source of the low-frequency time lag structure in blood oxygenation level-dependent signals

Toshihiko Aso[1,2,3]*, Shinnichi Urayama[3,4], Hidenao Fukuyama[3,4], Toshiya Murai[1]

**1** Department of Psychiatry, Kyoto University Graduate School of Medicine, Kyoto, Japan, **2** Laboratory for Brain Connectomics Imaging, RIKEN Center for Biosystems Dynamics Research, Kobe, Japan, **3** Human Brain Research Center, Kyoto University Graduate School of Medicine, Kyoto, Japan, **4** Research and Educational Unit of Leaders for Integrated Medical System, Center for the Promotion of Interdisciplinary Education and Research, Kyoto University, Kyoto, Japan

* aso.toshihiko@gmail.com

**Data Availability Statement:** The datasets analyzed for this study can be provided upon request. The Ethical Comittee of the Kyoto University Hospital does not permit public data

## Abstract

Perfusion-related information is reportedly embedded in the low-frequency component of a blood oxygen level-dependent (BOLD) functional magnetic resonance imaging (fMRI) signal. The blood-propagation pattern through the cerebral vascular tree is detected as an interregional lag variation of spontaneous low-frequency oscillations (sLFOs). Mapping of this lag, or phase, has been implicitly treated as a projection of the vascular tree structure onto real space. While accumulating evidence supports the biological significance of this signal component, the physiological basis of the "perfusion lag structure," a requirement for an integrative resting-state fMRI-signal model, is lacking. In this study, we conducted analyses furthering the hypothesis that the sLFO is not only largely of systemic origin, but also essentially intrinsic to blood, and hence behaves as a virtual tracer. By summing the small fluctuations of instantaneous phase differences between adjacent vascular regions, a velocity response to respiratory challenges was detected. Regarding the relationship to neurovascular coupling, the removal of the whole lag structure, which can be considered as an optimized global-signal regression, resulted in a reduction of inter-individual variance while preserving the fMRI response. Examination of the T2* and $S_0$, or non-BOLD, components of the fMRI signal revealed that the lag structure is deoxyhemoglobin dependent, while paradoxically presenting a signal-magnitude reduction in the venous side of the cerebral vasculature. These findings provide insight into the origin of BOLD sLFOs, suggesting that they are highly intrinsic to the circulating blood.

## Introduction

In functional magnetic resonance imaging (fMRI), there are 2 established physiological bases of signal change: neurovascular coupling (NVC) and autoregulation. The former involves a

sharing without explicit informed consent whether the data is from patients or healthy control participants. Our protocol did not include such an item and therefore sharing must be conducted in a form of collaborative work. Please contact the Comittee via Email: ethcom@kuhp.kyoto-u.ac.jp.

**Funding:** This study was supported by a Grant-in-Aid for Scientific Research on Innovative Areas (Non-linear Neuro-oscillology: Towards Integrative Understanding of Human Nature, JP15H05875) to TA and JP16H06395 and JP16H06397 to TM from the Japan Society for the Promotion of Science. The funders had no role in study design, data collection and analysis, decision to publish, or preparation of the manuscript.

**Competing interests:** The authors have declared that no competing interests exist.

local blood flow increase of 30–70% which gives rise to a 0.5–2% blood oxygenation level-dependent (BOLD) signal increase with around a 5-s delay [1]. This is the target phenomenon of fMRI as a tool for brain mapping, due to its limited spatial extent mainly involving local arterioles [2]. In contrast, autoregulatory responses in vessel diameter are found in a wide range of arteries including the internal carotid or middle cerebral arteries [3]. Detection of a compromised response in vascular disorders has proven useful for clinical purposes [4]. Importantly, there is no clear physiological distinction between these 2 phenomena as each involves multiple pathways [5]. Their traces in the fMRI signal are also uniformly postulated to reflect the increased cerebral blood flow and eventual dilution of deoxy-hemoglobin (Hb) in the postcapillary part of the vasculature, with the additional effect of a local blood volume increase [6].

In efforts to improve the efficiency of fMRI, studies have revealed various systemic physiological components in BOLD signal fluctuations. Physiological parameters, such as cardiac pulsation [7], blood pressure, and end-tidal carbon dioxide ($CO_2$) [8,9], are considered artifact sources. The contamination is expected to be emphasized in resting-state fMRI (rs-fMRI), where signals are evaluated without trial averaging [10]. However, discrimination between neuronal and non-neuronal components has been a major challenge due to the lack of validation techniques with spatial and temporal precision comparable to that of fMRI. Another source of difficulty is the fact that many neural and non-neural parameters are intercorrelated in this low-frequency range [8].

A focus of recent studies exploring this matter is the spontaneous BOLD low-frequency oscillation (sLFO, < 0.1 Hz) possibly encompassing multiple artifact sources [11,12]. One well-known sLFO source is the respiratory volume fluctuation involving a chemoreflex loop [13]. An sLFO in systemic blood pressure, known as the Traube–Hering–Mayer wave, has been shown to originate from another autonomic loop [14,15]. Moreover, associated sLFOs in blood flow and velocity have been found [16,17] and, later, transcranial Doppler ultrasonography and optical methods confirmed their traces within the brain [18,19]. This optically detected sLFO was found in both oxy- and deoxy-Hb with an interesting phase difference exclusively observed in the brain [20,21]. Additionally, sLFOs were found in electroencephalographic recordings, for which arterial vasomotion was suggested as the origin [22], although it is unclear how the vasomotion accounts for the fMRI signal mainly from the capillary bed.

As mentioned above, Hb-sLFOs are postulated to be of systemic origin [23–25]. Hence, it was not surprising to find a correlation between the global fMRI signal and extra-cerebral signals (near-infrared spectroscopy [NIRS] or MRI), but the constant time shift across body/brain parts was unexpected [26,27]. The similarity between the low-frequency phase map and perfusion MRI in healthy participants was a milestone in this direction, as it presented the perfusion time lag embedded in the BOLD signal [11]. The resilient nature of the lag map against the fMRI task condition was shown, further supporting its non-neuronal origin [28]. In parallel, a number of clinical studies have established the phase delays as a marker of cerebrovascular disorders [29–34]. Moreover, gross vascular anatomy has been detected consistently in these studies, replicating the results from respiratory challenges [35,36], which importantly suggests an equivalence between the sLFO and manipulated circulatory turbulences. Apart from patient data, a recent study involving healthy participants revealed changes in venous drainage patterns with normal aging [37]. A body of evidence thus empirically supports the biological significance of the low-frequency lag structure and its underlying principles.

The current analytical model of the BOLD lag structure assumes the presence of this signal variation from the very early stages of cerebral perfusion [38] (Fig 1A). Such synchronized variation should naturally affect the global mean signal, and the model is hence related to the unresolved fMRI global signal problem [39,40]. This view is not only compatible with the

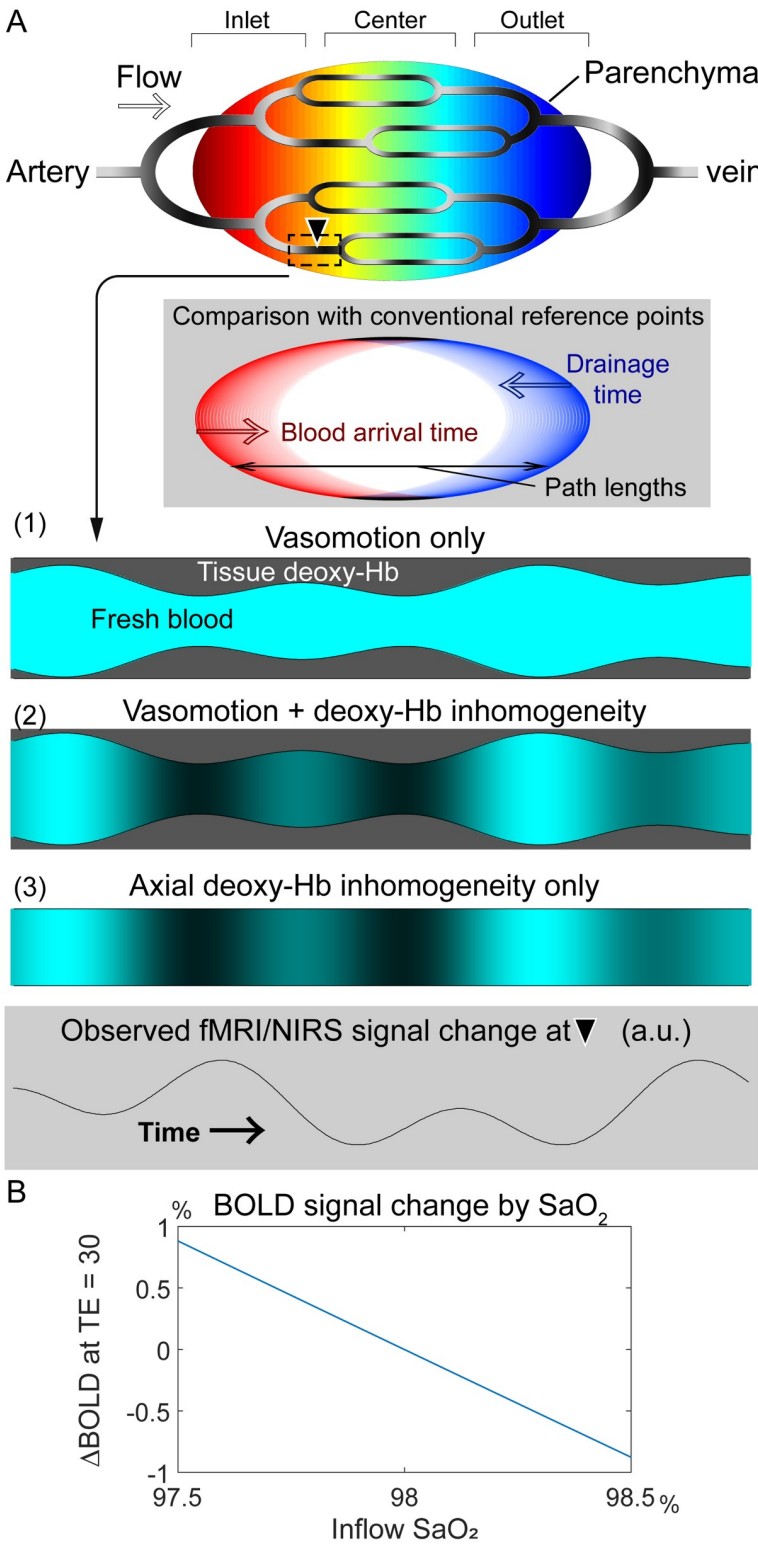

**Fig 1. A working model of BOLD lag mapping.** (A) A schematic of the cerebral vascular tree presenting possible physiological models that account for the fMRI and NIRS observations. Each brain voxel has its lag (phase) of the spontaneous low frequency oscillation (sLFO) relative to the global signal phase as the reference point, depicted in warm and cool colors. The inlet, center, and outlet parts of the gross vasculature are determined purely by the temporal relationship and not by vessel types; all signals should originate from the capillary bed. Another schematic (shaded

background) is inserted to illustrate the relationship between reference point selection and vascular path length, which accounts for the disagreement with other blood-tracking techniques. Below are 3 models with different physiological signal sources that are moving along the vasculature. The current model with constant deoxy-Hb requires vasomotion to account for the sLFO (1). While the axial inhomogeneity of the deoxy-Hb density may be linked with vasomotion (2), it can persist without it (3). The moving axial inhomogeneity creates a similar temporal profile with varying phases across regions. (B) Dependence of the BOLD signal on inflow oxygen saturation (SaO$_2$) at 3 T, TE = 30 ms. All other parameters were held constant at physiological values: oxygen extraction fraction = 45%, tissue blood volume fraction = 3%, and hematocrit = 40% (see Eqs (1) and (2)). BOLD, blood oxygen level-dependent; fMRI, functional magnetic resonance imaging; NIRS, near-infrared spectroscopy; TE, echo time; Hb, hemoglobin.

occasional favorable effect of global signal regression (GSR) in task fMRI [41] but has led some scientists to propose the removal of the lag structure as an approach for noise elimination [42–44]. Under this model, the elimination of the lag structure can be viewed as an optimized GSR (of the low-frequency component) for each voxel group. Conversely, with the presence of the BOLD lag structure, a simple GSR should retain a residual correlation between in-phase voxels that confounds fMRI analyses [43,45].

The fundamental and critical question remaining is the mechanism by which sLFOs (or respiratory maneuvers) create the BOLD lag structure. The BOLD response to neural activity via NVC is a well-documented passive process, involving the expansion of the intravascular compartment [46]. A typical model of the BOLD signal change is described as follows [47]:

$$\frac{\Delta BOLD}{BOLD_0} = M\left(1 - \left(\frac{CMRO_2}{CMRO_2|_0}\right)^{\beta}\left(\frac{CBV}{CBV_0}\right)\left(\frac{CBF}{CBF_0}\right)^{-\beta}\right), \tag{1}$$

where CMRO$_2$ stands for the cerebral metabolic rate of oxygen in a voxel and CBV/CBF represents the cerebral blood volume and flow, respectively. Beta (β) is an exponent of the power-law describing the relationship between T2$^*$ and the deoxy-Hb amount that only depends on CMRO$_2$ and CBF in a reciprocal manner, under the assumption of a negligible inflow of deoxy-Hb. M is the factor for the BOLD susceptibility effect, defined as:

$$M \equiv TE \cdot B_0 \cdot CBV_0 \cdot [deoxy - Hb]_{V0}^{\beta}, \tag{2}$$

where B$_0$ is the main magnetic field strength and TE represents the echo time. Triggered by vasodilation of the arteriole, this effect is diminished by an inflow of fresh blood, which increases the MR signal. This baseline BOLD effect has been modeled in the formula for the off-resonance frequency shift (δω) as follows [48,49]:

$$\delta\omega = \gamma \cdot \frac{4}{3} \cdot \pi \cdot \Delta\chi_0 \cdot Hct \cdot OEF \cdot B_0, \tag{3}$$

where γ is the gyromagnetic ratio (42.58 MHz/T), $\Delta\chi_0$ is the susceptibility difference between the fully oxygenated and fully deoxygenated blood, and Hct is the hematocrit (volume fraction of erythrocytes to the blood volume, typically around 40%). The oxygen extraction fraction (OEF) represents the only source of deoxy-Hb under the assumption of 100% oxygen saturation (SaO$_2$) in the inflow.

Variations of this base susceptibility can occur due to the local Hct and SaO$_2$ changes and, in fact, have been shown to cause intersession variabilities [50,51]; however, within-session fluctuations have rarely been considered [52]. For example, even at a constant Hb density and oxygen partial pressure, CO$_2$ fluctuation, a driving factor of vasomotion, alone can modify SaO$_2$ through pH changes [53]. Although the assumption of constant base deoxy-Hb concentration may be sufficient for modeling its dilution by NVC [54], other effects might not be negligible in non-neuronal fluctuations (Fig 1b).

In this exploratory study, we conducted 3 investigations to further advance our knowledge on the BOLD lag structure and its underlying physiology. We first evaluated if changes in blood transit velocity are embedded in the BOLD low-frequency phase to confirm its behavior as a virtual tracer [a preliminary analysis of these data was presented previously as a poster [55]]. Next, we investigated the effect of eliminating the lag structure from task-based fMRI, which had not been tested previously. Finally, we used multi-echo imaging to assess the components of the BOLD signal that determine the lag structure. One of the recent approaches toward fMRI denoising has focused on $S_0$ fluctuations (signal at TE = 0, which is the baseline MR signal from the fluid compartments), as it is weakly associated with neural activity [56–59]. In contrast, the total-Hb sLFO, detected by NIRS, is interpreted as a local CBV change under the assumption of a constant Hct, which should also affect the non-BOLD component via changes in plasma volume and inflow [60]. Notably, the contributions of the $T2^*$ and $S_0$ components may differ from NVC contributions to the BOLD lag structure; hence their impact may have been overlooked in studies based on trial averaging. From the influences of 3 different fMRI task paradigms, including a simple reaction-time visuomotor task, a short breath-holding task, and a hyperventilation task on the neural and non-neural components of the fMRI signal, we sought further validation of our hypothetical model of the BOLD lag structure.

## Materials and methods

### Participants and experimental procedures

Twenty-one healthy participants (8 women, 19–26 years of age) participated in Experiment 1; only 1 person was excluded from the analysis because of an abrupt head motion, which prevented BOLD lag mapping (see Data processing). The remaining 20 participants performed the sparse visuomotor and 10-s breath-holding tasks, but the hyperventilation task was only performed by 18 participants, as it was introduced after the first 2 individuals had concluded their participation. Another 21 participants were recruited for Experiment 2, involving multi-echo acquisition, all of whom performed the above 3 tasks but with an 18-s version of the breath-holding task. To avoid vigilance level fluctuations, all MRI sessions were scheduled in the morning and the participants were encouraged to sleep well the previous night.

The protocol for this study was approved by the internal ethics review board of Kyoto university. The participants provided written informed consent in advance, according to the Declaration of Helsinki, for the analysis of anonymized MRI scans and simultaneously acquired physiological data.

### Image acquisition

A Tim-Trio 3-Tesla scanner (Siemens, Erlangen, Germany) with a 32-channel phased-array head coil was used for MRI acquisition. For Experiment 1, $T2^*$-weighted echo-planar images were acquired using multiband gradient-echo echo-planar imaging (EPI) [61] with the following parameters to cover the entire cerebrum: $64 \times 64$ pixels, 35-slice interleave, 192-mm field of view (FOV), 3.5-mm slice thickness, repetition time (TR)/ TE = 500/35 ms, flip angle = 40˚, and a multiband factor of 5. Three 9-min runs (1,080 volumes) were acquired for each of the 3 task conditions. The same pulse sequence program was used in Experiment 2 but with multi-echo settings: TE1 = 7.76 ms and TE2 = 25.82 ms for the first 6 participants and TE1 = 11.2 ms and TE2 = 32.78 ms for the remaining 15 participants. A smaller multiband factor of 2 was selected to allow for the short TE in combination with parallel imaging using GeneRalized Autocalibrating Partial Parallel Acquisition. Other acquisition parameters were: TR = 1,300 ms, flip angle = 65˚; 36-slice interleave, FOV = $256 \times 192$ mm$^2$, $64 \times 48$ matrix, and 3.5-mm

slice thickness. Three 7-min (323 TR) runs were acquired. Seven participants in Experiment 2 underwent 2 additional runs with a shorter TR of 700 ms and a flip angle of 45˚ to examine the sensitivity of the respiration-related signal component to inflow modulation. At the end of every experimental session, a 3-dimensional (3D) magnetization-prepared rapid acquisition with gradient echo (MPRAGE) T1-weighted image was acquired for obtaining anatomical information [28]. A dual-echo gradient-echo dataset for $B_0$ field mapping was also acquired after the BOLD scan in the same orientation.

## Task conditions

Throughout the experimental session, task instructions were presented via a liquid crystal display (LCD) monitor inside the scanner room, viewed through a mirror. Beat-to-beat fluctuations in the mean arterial pressure and heart rate were obtained via a non-invasive MR-compatible device (Caretaker, BIOPAC Systems, Inc., Goleta, CA, USA). Careful instructions were provided to the participants on how to avoid motion, especially during the respiratory challenges.

**Sparse visuomotor task.** A simple visuomotor task with a varying intertrial interval of 6 to 24 s was performed during the first run. Participants were instructed to press a button with their right index finger as soon as the computer screen changed from "Please hold still" to "Press the button." The screen returned to "Please hold still" at the button press or after 3 s, if the participant had not pressed the button.

**Breath-holding task.** To minimize head motion induced by the tasks, the 2 respiratory challenges were adapted to be less strenuous than in earlier studies. In Experiment 1, the breath-holding task was cued by a "Hold your breath" instruction on the screen, at which point the participants were asked to immediately hold their breath, irrespective of the respiration phase. The holding periods lasted 10 s and were separated by 90-s intervals. This short duration was selected to minimize strain that can cause body movements, while evoking a detectable autoregulatory response [4]. In Experiment 2, a longer holding period of 18 s after a brief inhalation for 2 s was used to evoke a more pronounced vasodilation.

**Hyperventilation task.** The hyperventilation task involved paced breathing at 0.2 Hz for 25 s, separated by a 30-s rest. Each 5-s cycle began with the screen presenting "Please inhale" for 1.5 s, followed by "Please exhale slowly," lasting 3.5 s. Participants were instructed to breathe as deeply as possible, while avoiding head movement. A short inspiration period was selected to suppress motion by minimizing movements in the thoracic cage and spine.

## Data processing

For image processing, SPM12 (Wellcome Department of Cognitive Neurology, London, United Kingdom) and FSL5 (FMRIB Software Library, www.fmrib.ox.ac.uk/fsl) [62] were used in combination with in-house MATLAB scripts. Off-resonance geometric distortions in the EPI data were corrected using FUGUE/FSL with $B_0$ field maps. After inter-scan slice-timing correction, head motion was compensated by 3D motion correction and data repair [63]. The repairing procedure aimed to remove motion-related signal dropout and involved searching for time points satisfying 2 stringent criteria: (1) global signal changes between consecutive volumes exceeding 1% and (2) head displacement exceeding a Euclidian distance of ± 1 mm or ± 1˚ rotation per TR. The affected time points were replaced with linearly interpolated values, but this procedure was required in only 6 of the 41 participants.

The data were further cleaned by regressing out 24 head motion-related parameters. Unlike in previous studies, the 6 rigid-body parameter time series were not directly used because of the possible contamination of the motion parameters with the global signal when the

participants were immobile [64]. We used the first temporal derivatives of the motion parameters, their versions after being shifted by 1 TR, and the squares of those 12 time series [65]. Images were spatially normalized to the template space using the T1-weighted anatomical image and resliced to a 4-mm isotropic voxel size to achieve a high voxel temporal signal-to-noise ratio.

**Lag mapping.** A recursive technique was used [28] after temporal bandpass filtering at 0.008–0.07 Hz to ensure that the phase was uniquely determined within the cross-correlation range. Whereas lag was tracked up to 7 s for most analyses, it was limited to 4 s in both directions, upstream and downstream, for the calculation of the relative BOLD transit time (rBTT, see below). This shorter tracking range allowed a higher cutoff frequency (0.12 Hz) to preserve the high-frequency component in the velocity change profiles.

The global mean signal was used to select the initial seed that defined the reference phase (lag = 0). First, voxels presenting a cross-correlogram peak at 0 with the global signal were determined. The time course averaged over this set of voxels served as the initial reference. In each step of the recursive procedure toward up- and downstream, a cross-correlogram was calculated between the time series obtained from the previous seed voxels and every undetermined voxel to find a set of voxels with a peak at ± 0.5 s, which then served as the new seed. This tracking part retained some voxels without any lag values because cross-correlogram peaks < 0.3 were not used following earlier works [11]. These voxels were later filled 1 by 1 with average phases from voxels with similar time courses and correlation coefficients > 0.3. There were single isolated holes even after this procedure, which were filled by linear interpolation, using the 6 neighbors. There is a concern that this correlation coefficient threshold is too low to accurately claim a significant correlation. However, recursive tracking involves finding the cross-correlogram peak precisely at ± 0.5 s, which conveys different information from its height. Besides, most earlier works empirically supporting the biological significance of this phenomenon involved no thresholding. When the threshold correlation coefficient was increased to 0.6, most brain voxels required the hole-filling procedure, but we still obtained lag maps (by between-voxel intra-class correlation $(2,1) > 0.4$) in 16 out of 20 participants during resting state and 17 during 10-s breath holding.

**fMRI analysis on "cleaned" BOLD datasets.** Individual BOLD data from Experiment 1, after the above pre-processing steps including the motion parameter regression, served as the reference or "raw" dataset. GSR with a low-pass filtered global signal, a normal GSR, global scaling implemented in SPM12, and without the perfusion lag structure ("deperfusioned") were compared with the raw dataset.

The GSR involves regression by the global signal and extracting the residuals in each voxel. In deperfusioning, instead of the uniform regressor, a corresponding time series was used for each voxel group by the lag value ("dynamic" GSR) [43]. For global scaling, the raw dataset was entered into the same SPM pipeline, except for the option of internally dividing each volume by its global mean signal instead of the constant session mean. The normal GSR and global scaling thus affected all frequency ranges, whereas the GSR and deperfusioning removed only the low-frequency components.

Random effects analysis [66] was used to evaluate the effect of these procedures on the fMRI results from the standard analysis framework. For the visuomotor task, the neural response to each trial was modeled as an event of 0.5 s in duration. For the hyperventilation condition, a boxcar with a 25-s duration modeled the activation related to volitional respiratory control. Similarly, both the onset and offset timing of breath holding were used to model the time-locked neural activity. The canonical hemodynamic response function was convolved to the model time series to create the regressors of interest. The threshold for all activation

maps was p = 0.05 after correcting for multiple comparisons, using family-wise error across the whole brain [67].

**Relative BOLD transit time.** This analysis was performed on data from Experiment 1, acquired at a short TR of 0.5 s. The lag structure consisted of a lag map and the set of time series averaged over the voxels with the same phase. This structure represents the propagation of the sLFO phase along the vessels. The phase is expected to move across adjacent regions of the vascular tree every 0.5 s, the lag tracking step, on average. However, if there is variation in propagation velocity, there would be a deviation of the instantaneous phase difference from 0.5. Here, the phase difference is supposed to reflect the time the blood requires to cross the boundary between the neighboring voxel groups.

Based on this supposition, the phase difference fluctuation was calculated from each of the 16 pairs of seeds corresponding to lags of -4 to +4 s at 0.5-s intervals. Due to the broad frequency range of the fluctuations of interest, we chose a smooth sliding window algorithm (window length = 30 s, Kaiser window with a β value of 4) instead of using analytic methods (e.g., Hilbert transform). The region time series were resampled to a 0.02-s resolution to capture minute phase difference fluctuations (4%) from 0.5 s. Instantaneous phase differences from each pair of seeds were averaged over respiratory task events and divided by 0.5 to obtain the time course of the rBTT. The rBTTs from the 16 pairs of neighboring regions were then averaged to obtain the regional or global rBTT. According to this model, the inverse of the rBTT corresponded to the instantaneous velocity (relative to the baseline average velocity), as the rBTT should reflect the average time required to traverse fixed distances.

**Multi-echo combination.** For Experiment 2, involving multi-echo acquisition, $S_0$ and $T2^*$ datasets were created by a simple estimation used in earlier works [57,59]. We assumed a single compartment monoexponential decay of the MR signal as follows:

$$S(TE) = S_0 \, exp(-TE/T_2{}^*), \tag{4}$$

where $T2^*$ and $S_0$ were calculated for each TR as follows [56,57]:

$$T2^* = (TE2 - TE1)/\ln(S_1/S_2), \tag{5}$$

and

$$S_0 = S_1{}^{TE2/(TE2-TE1)}/S_2{}^{TE1/(TE2-TE1)}, \tag{6}$$

where $S_1$ and $S_2$ are the acquired signals at TE1 and TE2, respectively. Negative or $T2^*$ values exceeding 100 ms were considered as noise and were ignored. All four datasets ($S_0$, $S_1$, $S_2$ or BOLD, and $T2^*$) were entered into the same analysis pipeline used for Experiment 1 while accounting for the different TR value of 1.3 s. The following analyses were performed on a resampled time course with a TR of 0.5 s.

The interaction between the signal component and vascular anatomy was examined by extracting the signal time series from the inlet, center, and outlet parts of the vascular tree, based on the individual lag map created from the BOLD (i.e., at TE2) dataset. By exploiting the longer lag tracking range (± 7 s) than the one used in the rBTT analysis in Experiment 1, the center region in this analysis covered a wider range (± 2.5 s). Using the JMP12 software (SAS Institute, Cary, NC), the magnitude and phase of the regional signals were analyzed by repeated-measures ANOVA followed by post hoc analyses, using Tukey's honestly significant difference (HSD) test. Statistical significance was set at $p < 0.05$. Additional analyses were performed to examine the origins of the signal components, including a region-of-interest analysis and an SPM analysis of respiration phase-related small $S_0$ fluctuations.

## Results

The average root mean square of the head motion was measured as the framewise displacement (i.e., the shift in the position of the brain in 1 volume compared to the previous volume), was 0.039 ± 0.007 mm (mean ± standard deviation (SD) over participants) for Experiment 1, with a maximum displacement of 0.37 ± 0.18 mm, and 0.034 ± 0.015 mm for Experiment 2, with a maximum of 0.24 ± 0.12 mm [68].

Fig 2A shows an individual and the average BOLD lag maps during the sparse visuomotor task. Warm colors indicate a positive travel time from those voxels to the phase of the global LFO, signifying that the voxels are considered "upstream." Most brain voxels fell into the -4 to +4 s range (mean ± SD, 93.8 ± 2.9%). The recursively defined seed time courses are shown in Fig 2B with warm colors indicating averaged time series from the upstream voxels. This time course of the lag structure was poorly correlated with the visuomotor task (white vertical lines) or the evoked NVC response (trace in dark gray), supporting a non-neuronal origin of the sLFO. This was confirmed by the correlation coefficient between the global mean signal and the modeled response that was not significantly different from 0 (0.0265 ± 0.140, mean and SD

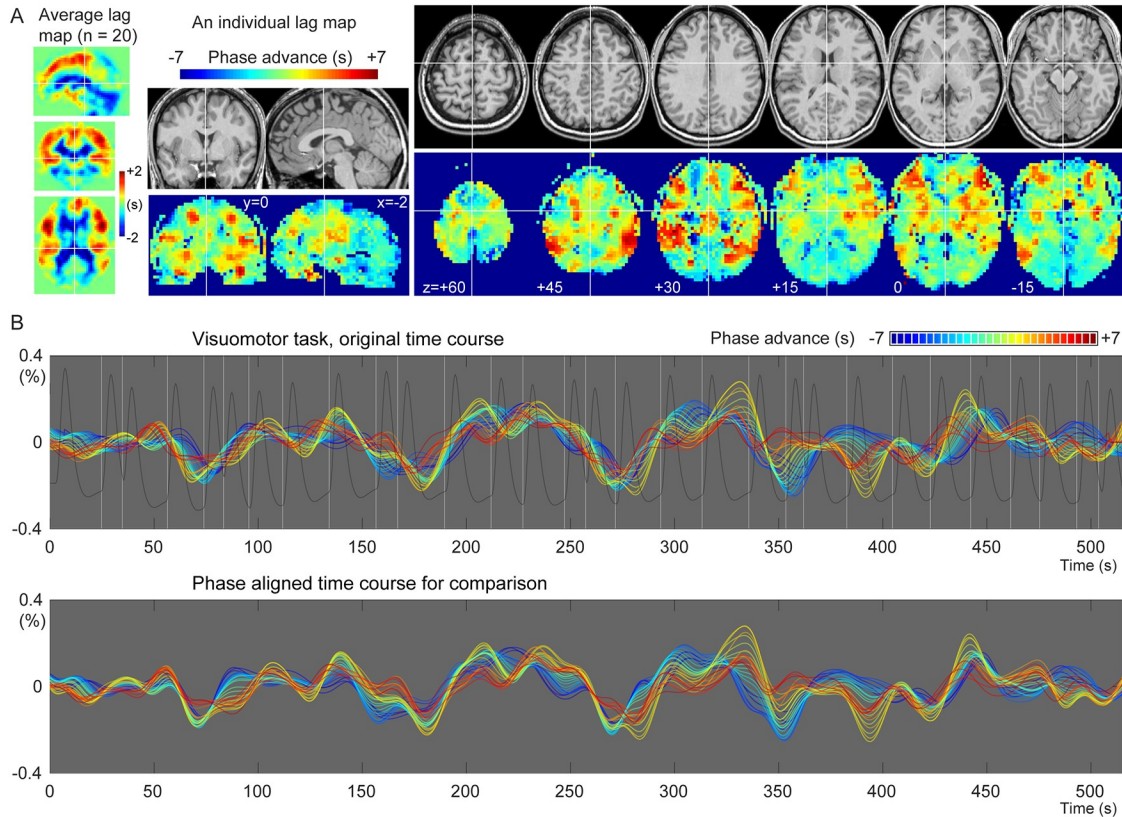

**Fig 2. Representative spatial and temporal profiles of the BOLD lag structure.** (A) A lag map created by tracking the BOLD sLFO phase up to 7 s toward both up- and downstream is shown. The group average map shows the gross vascular anatomy with the early phase (positive values) distributed in the middle cerebral artery territories. The individual map provides more detailed information. (B) Seed time courses updated at each step (0.5 s) of the recursive lag tracking procedure, representing the temporal aspect of the lag structure, are shown. The warm-colored traces with advanced phases originate from the voxels with the same colors in the lag map, corresponding to the inlet or arterial side of the vasculature. A gradual change in the temporal profile is noted on top of the slow component, which is stable across regions. The white vertical lines indicate the timing of the visuomotor task, whose NVC was modeled by the hemodynamic response function (dark gray trace). The detected LFO was poorly correlated with the task-related fluctuation of neuronal origin (see main text). BOLD, blood oxygen level-dependent; NVC, neurovascular coupling.

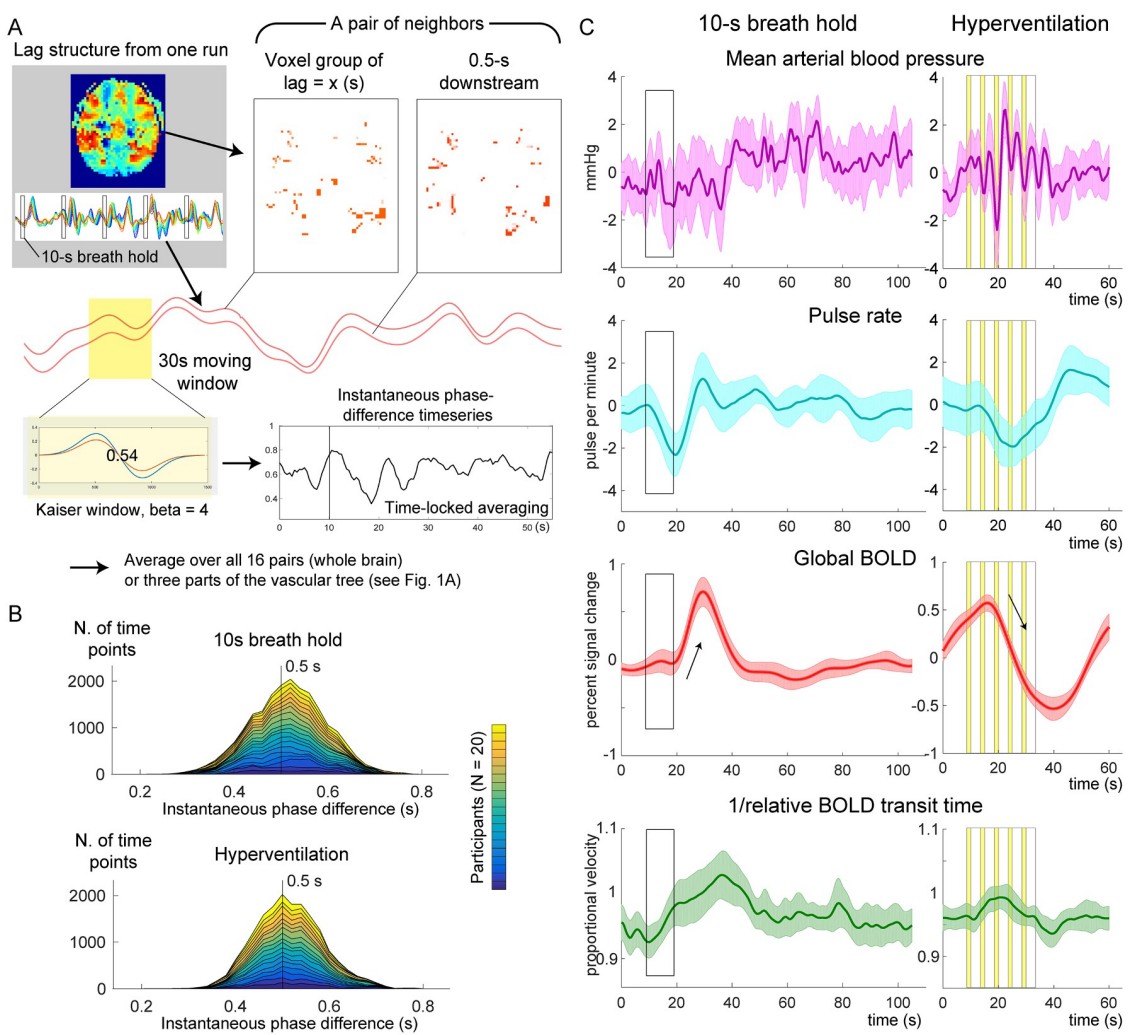

**Fig 3. Measurements of the relative BOLD transit time (rBTT).** (A) A schematic illustration of the analysis is presented. For each pair of voxel groups representing neighboring regions of the vascular tree, the 2 average time series were fed into a moving window analysis of instantaneous phase difference. (B) Time point histograms of the instantaneous phase difference averaged over 16 pairs of neighboring regions in the lag structure are shown. Twenty participants are stacked. Deviation of the phase difference from 0.5 s reflects a fluctuation of the flow velocity, although the slight shift of the peak is observed in the breath hold data presumably due to windowing. (C) Responses to the respiratory challenges of the physiological recordings, raw BOLD signal, and inverse of the rBTT that can be interpreted as relative velocity are shown. Thin arrows indicate the autoregulatory response of the raw BOLD signal. The 5 cycles of volitional breathing for hyperventilation are indicated with yellow bands representing the inspiration phase. Shaded areas indicate the 95% confidence interval of the mean across participants. BOLD, blood oxygen level-dependent.

over 20 participants), although still slightly positive in some participants and its regression affected the activation maps (see section 3.2). In addition to the constant phase shift across regions, the lag structure time courses presented minute fluctuations of the phase difference (i.e., the temporal relationship between the lines) over time.

## Flow velocity information in the instantaneous phase

A transient change in the propagation velocity of the low-frequency phase, obtained as the inverse of the global rBTT, was found in response to the respiratory challenges. Fig 3B shows the stacked histograms of the distribution of the instantaneous phase difference measured by the moving window analysis. This instantaneous phase difference is interpreted as the time the

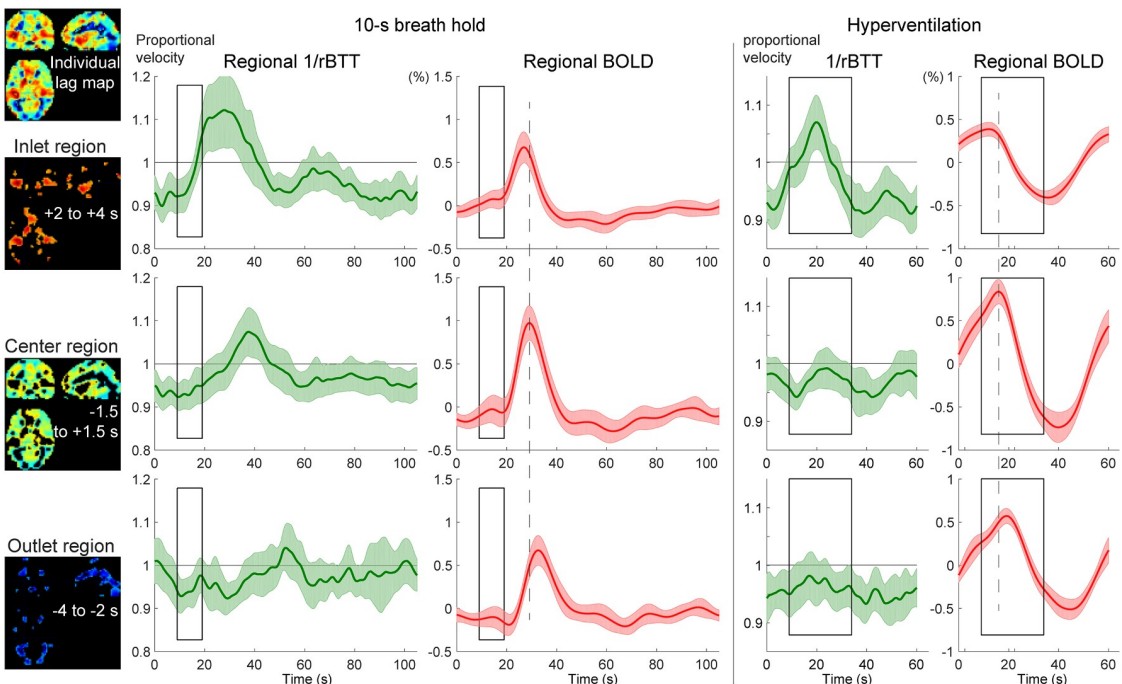

**Fig 4. Autoregulatory responses to respiratory challenges in the 3 vascular regions, using the same conventions as in Fig 3.** The inlet (arterial side), center, and outlet (venous side) of the parenchymal vasculature were defined for each participant, using the lag map created from the session data. The relative changes in propagation velocity were detected as the inverse of the instantaneous transit time deviation from 0.5 s (rBTT), since the lag mapping was performed in 0.5-s increments. The BOLD signal time courses present different peak latencies for the 3 regions (broken lines are aligned to the center region peak), directly reflecting the lag structure, but with similar profiles, ruling out its effects on the rBTT measurement. Shaded regions represent the 95% confidence intervals (N = 20). BOLD, blood oxygen level-dependent.

blood takes to move over a unit distance that requires 0.5 s on average. The time courses are presented in Fig 3C. After approximately 10 s of delay, an increase and a decrease of the BOLD signal was found during breath holding and hyperventilation, respectively (arrows).

The temporal profile of the instantaneous velocity (green curves) was roughly in phase with the global BOLD response but had different onset and peak timings, indicating different physiological bases. To evaluate the relationship of this phenomenon with the vascular structure, the inlet, center, and outlet regions were separately analyzed (Fig 4). In both respiratory challenges, there was a clear asymmetry over the vasculature, with pronounced velocity responses in the inlet region. Despite similar BOLD response profiles, the velocity response was not clearly found in the outlet region.

## Effect of deperfusioning on the detection of neurovascular coupling

Removal of the whole lag structure exhibited unique effects on the fMRI analysis. While the mean and SD images from the individual activation maps showed similar spatial distributions, all procedures reduced the sensitivity compared to the raw dataset where only the motion-related variances were removed (Fig 5). However, after the deperfusioning procedure, primary motor cortex activation was successfully detected by decreased interindividual variances (black circles). The effect of these procedures was not uniform between the respiratory challenges (Fig 6), but some interpretable clusters were selectively captured by the "deperfusioned" signals despite the reduced cluster number. For example, the laryngeal motor cortices were detected at the onset and offset of the 10-s breath-holding sessions [69]. In the hyperventilation

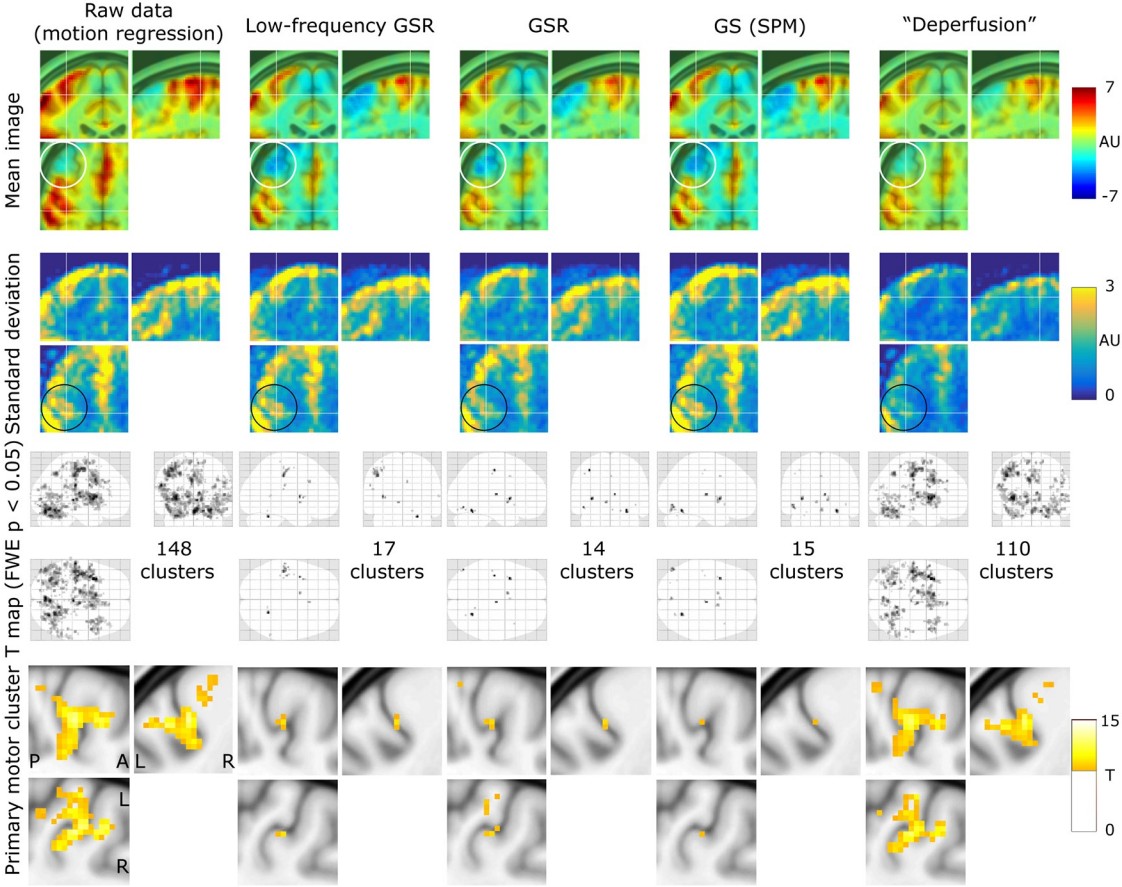

**Fig 5. The effects of the denoising procedures on the detection of neurovascular coupling during the visuomotor task.** Mean and standard deviation maps from 20 individual activation maps (contrast images) illustrate the spurious negative responses by the global-signal based methods (white circle) and the reduced between-participant variation by the lag structure removal or deperfusioning (black circle). The threshold for the activation maps was p = 0.05, corrected for FWE of multiple comparisons over the entire brain, and zoomed in to show the "hand-knob" of the left primary sensorimotor cortex. FWE, family-wise error corrected; GSR, global signal regression; GS, global scaling; SPM, SPM12 software.

condition, bilateral recruitment of the putamen was noted. Additionally, a premotor peak was found at coordinates [+56, 0, 40].

Voxel histograms from the group SPM analyses showed clear leftward shifts by the global signal removal, indicating spurious deactivations (Fig 7). The correlation of the neuronal response with the global signal (or the extracted sLFO) was near 0, as described above, but it varied across participants and tended to be positive. This trace of NVC may have created the spurious deactivation after regression. Notably, this effect was very weak after deperfusioning across all 3 conditions (green plots), despite a large amount of variance removed by the procedure.

## Magnetic resonance signal components of the lag structure

As depicted in Fig 8, the percent signal change of the sLFO, or the lag structure amplitude, revealed a clear $T2^*$-dependence with a significant reduction of amplitude in the outlet (i.e., the venous side of the gross vasculature [post hoc Tukey's HSD between $T2^*$ signals from the inlet and outlet regions, following a repeated-measures ANOVA]).

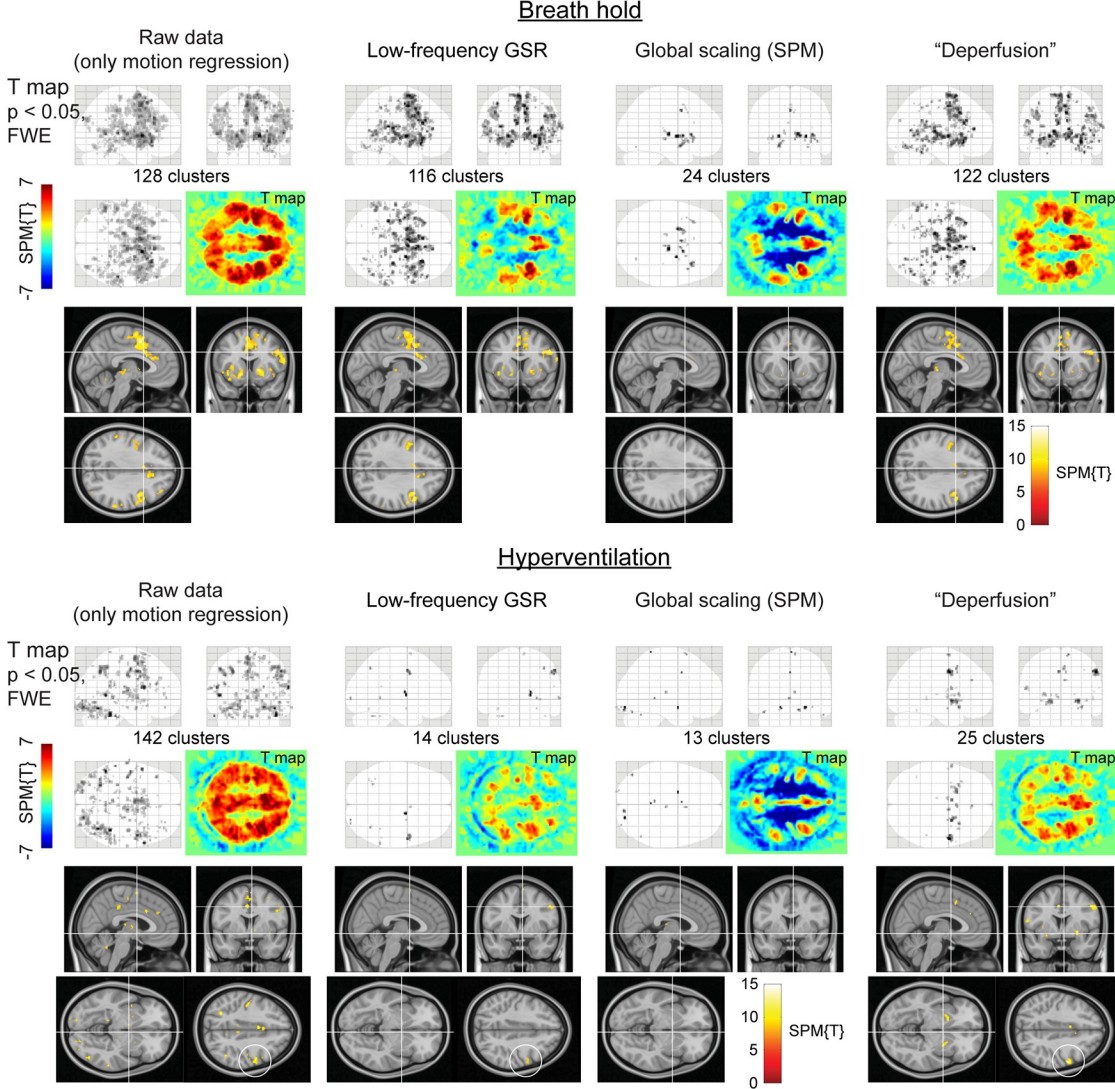

**Fig 6. SPM results of the respiratory challenges.** Color panels show un-thresholded t map slices at the height of the laryngeal motor cortex (z = 35), revealing spurious negative responses (cool colors) after global signal removal but not after the deperfusioning procedure. In the hyperventilation condition, the "deperfusioned" data revealed clusters in the bilateral putamen and premotor cortex (white circle), consistently with the findings of earlier reports. FWE, family-wise error corrected; GSR, global signal regression; GS, global scaling; SPM, SPM12 software.

In lag maps from the 3 $T_2^*$-weighted images, there were some effects of the respiratory challenges, but the gross cerebral vascular structure was preserved across tasks; the periventricular regions and major venous sinuses were uniformly found downstream (i.e., with negative arrival time) of the global signal phase, while the cortical territory of the middle cerebral arteries exhibited earlier arrival (Fig 9A). Only the $S_0$ image presented a different lag structure, according to the image similarity (Fig 9A, right panels).

Temporal analysis of the signal components revealed significant main effects of both region [$F (2,532) = 16.877$, $p < 10^{-6}$] and $T_2^*$ weighting [$F (2,532) = 280.786$, $p < 10^{-6}$] (S1A Fig). The $S_0$ time series failed to show a correlation with the $T_2^*$-weighted signals, but the z-value in the inlet (i.e., the arterial side) differed significantly from that in other regions ($p < 0.05$, Tukey's HSD). A region effect was also found for phase relationships (S1B Fig). Similarly, the

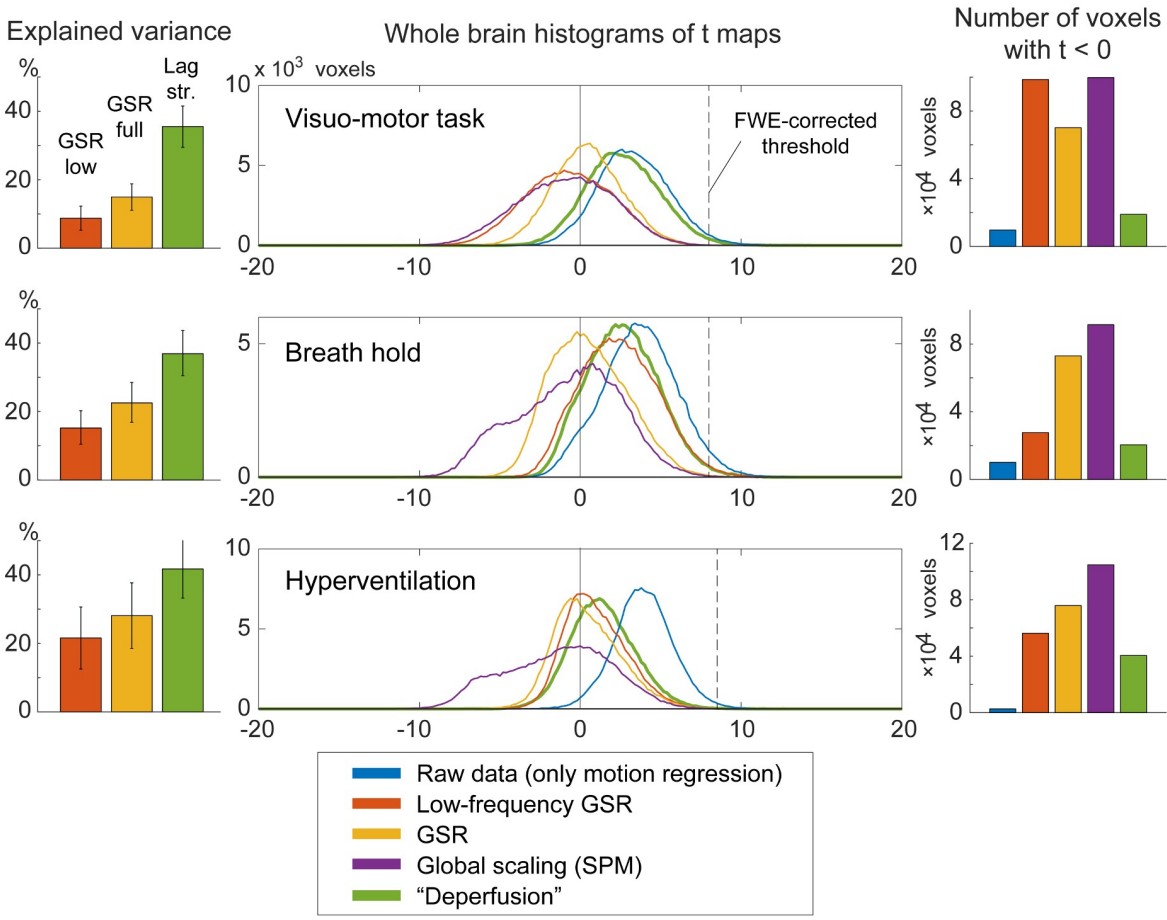

**Fig 7. The effects of preprocessing on group SPM analyses.** Removal of the lag structure or deperfusing resulted in the greatest reduction of the signal variance under all 3 tasks (left panels). Nevertheless, the spurious negative task responses were attenuated in comparison to the removal of global fluctuation (middle and right panels). Error bars indicate the standard deviation. The vertical broken line in the histogram indicates the statistical height threshold of p = 0.05, corrected for multiple comparisons by the family-wise error (FWE) rate. GSR, global signal regression; SPM, SPM12 software.

main effects for both region [F (2,532) = 27.548, $p < 10^{-6}$] and T2* weighting [F (2,532) = 44.679, $p < 10^{-6}$], as well as their interaction [F (4,532) = 24.902, $p < 10^{-6}$], were significant. The phase of the TE1 signal, which is less T2*-weighted than that of the BOLD signal, gradually advanced, finally showing a phase lead in the outlet region, further supporting an interaction between the signal components and vascular regions. These signal phase dissociations within regions are displayed in S1C Fig. Significant differences among the 3 T2*-weighted signals were also found after the post-hoc test (p < 0.05). The signal-region interaction was evident in the signal response to the respiratory challenges shown in Fig 9B. Note that the traces contain higher frequency components that were eliminated prior to lag mapping. In contrast to the changes in T2* responses for both phase and magnitude, the $S_0$ component was stable across vascular regions.

To further investigate the signal origin, we extracted the response in the motor/premotor area activation peak, where NVC was expected to dominate (S2A Fig). For this analysis, the group activation map from Experiment 1 was used to define the regions-of-interest in order to avoid bias. We found high-frequency components dominating the $S_0$ responses in comparison to the responses from larger regions shown in Fig 9B. During hyperventilation, respiratory

## Mean (Magnitude) vs. Region

### → T2*-weighted

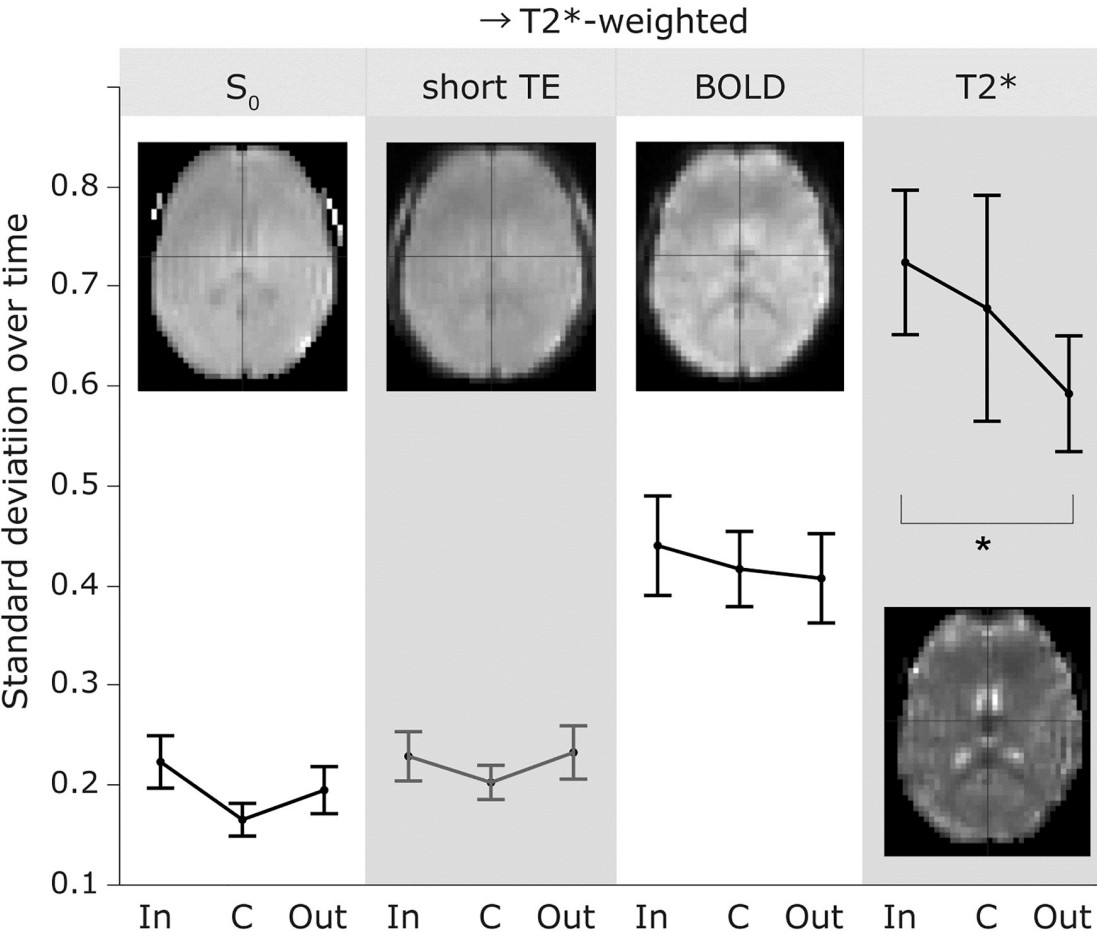

**Fig 8. The standard deviation of the percent signal change as a measure of sLFO magnitude.** No main effect of vascular region was observed by repeated-measures ANOVA, but the $T2^*$ magnitude was significantly different between the inlet and outlet sides ($p < 0.05$, Tukey HSD). The short-TE image was not included in the ANOVA to avoid data redundancy. Each error bar is constructed using a 95% confidence interval of the mean. sLFO, spontaneous low-frequency oscillation; ANOVA, analysis of variance; TE, echo time; In, inlet; C, center; Out, outlet.

phase-related fluctuations were observed with a signal decrease initiated by inhalation and followed by a positive deflection during exhalation. The $T2^*$ signal also exhibited small fluctuations but with a different phase, possibly dominated by the NVC component.

We conducted an additional analysis for the fast respiration-related non-BOLD components. The spatial distribution of this response is shown in S2B Fig. Interestingly, there was a clear anterior-posterior segmentation of response polarity, with the posterior regions presenting the opposite phase of the fast $S_0$ deflection by respiratory maneuvers. Some additional symmetrical structures were found in deep-brain regions, near the deep middle cerebral and inferior ventricular veins, implying a unique vascular involvement. Importantly, this spatial pattern was not that of typical motion artifacts that can accompany volitional respiration in spite of the careful instruction.

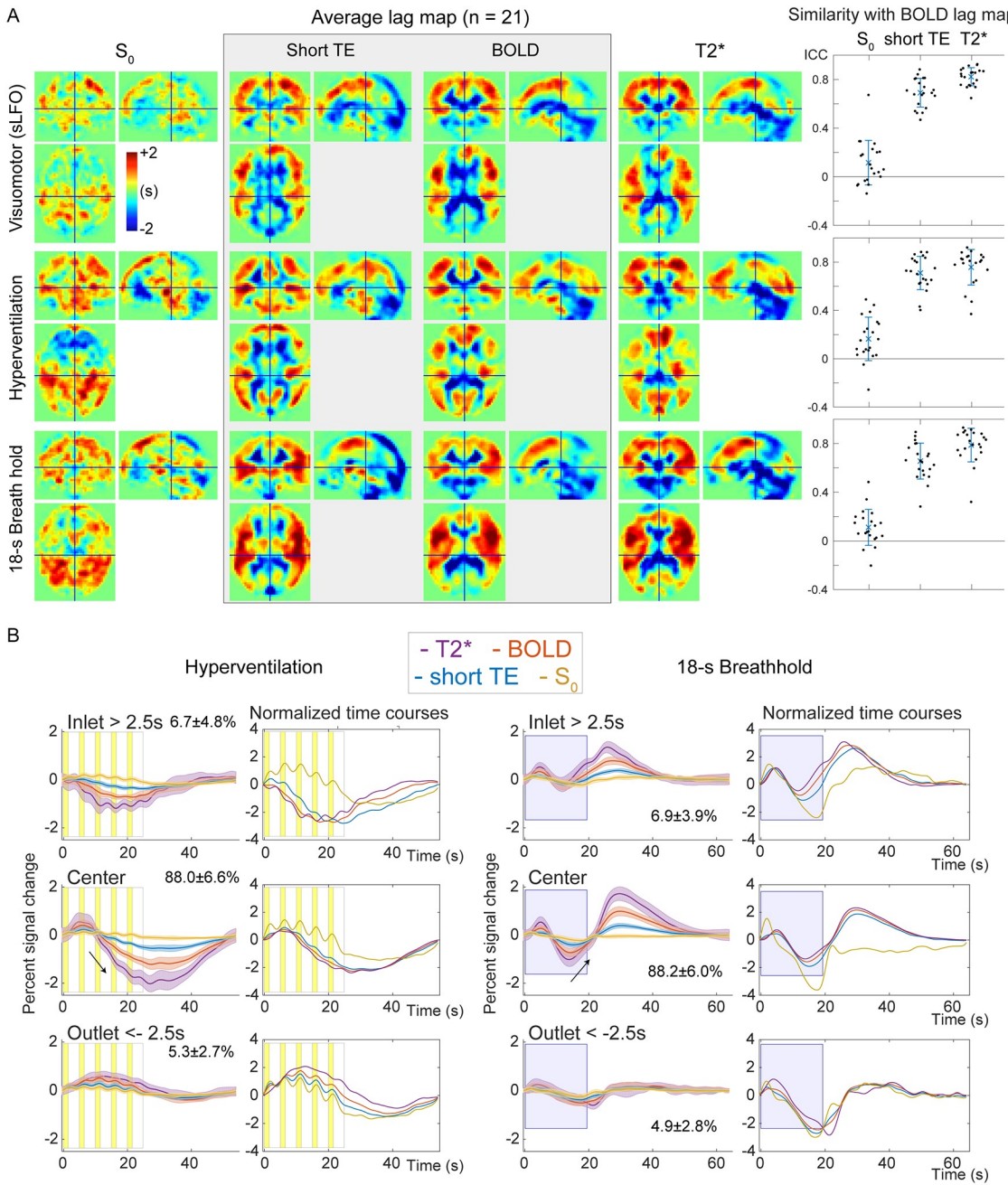

**Fig 9. Analysis of the signal components and the lag structure.** (A) $S_0$ and $T2^*$ signals were interpolated based on multi-echo acquisitions at short and typical TEs for BOLD fMRI. The lag map created from the $S_0$ image shows a unique structure but fails to reflect the arterial and venous structures that are consistently found in the BOLD lag map. Using the $T2^*$-weighted signals, the lag map changes upon respiratory challenges, which should primarily reflect the modification of the perfusion pattern; however, an interaction with the signal component is not excluded. The right panels show intraclass correlation coefficients as a quantitative measure of within-participant image similarity with the BOLD lag map, with error bars indicating 95% confidence intervals. (B) Temporal profiles from the 3 vascular regions indicate an absence of region effects in the $S_0$ signal, suggesting a globally uniform mechanism underlying the $S_0$ response. BOLD, blood oxygen level-dependent; ICC, interclass coefficient; TE, echo time; fMRI, functional magnetic resonance imaging; sLFO, spontaneous low-frequency oscillation.

Finally, S2C Fig shows the data from a subset of participants, obtained using a different TR/flip-angle setting to manipulate the inflow effect. The T2* response was smaller than that shown in Fig 2A, presumably due to the short TR. The different TR also contributed to the rich high-frequency components by the fast sampling rate. The slow $S_0$ change was also diminished, but the respiration-related fast component was relatively preserved suggesting the absence of a strong inflow effect.

## Discussion

The principal findings of this study are summarized as follows. First, based on the instantaneous phase difference within the BOLD lag structure, we observed a small blood flow velocity change selectively in the inlet region of the vasculature. Next, the complete elimination of the lag structure reduced interindividual variance and spurious deactivation, supporting our hypothesis that NVC could be observed more specifically by this deperfusioning procedure. This finding is in agreement with the results of earlier work on resting-state fMRI [43]. Finally, the lag structures in the $S_0$ (or non-BOLD) component did not correlate with that from T2*, either spatially or temporally. We also found a vascular region-dependent change in the T2* sLFO, with a decreased amplitude in the outlet part close to major veins, in contrast to the $S_0$ response that remained constant; this finding replicates a previous observation in the raw BOLD signal [28]. The $S_0$ component exhibited a unique brain region-dependent response to the respiratory phase, suggesting that certain perfusion parameters specifically contribute to this component, but not the perfusion lag. Overall, the BOLD low-frequency phase behaved as a deoxy-Hb-based virtual contrast agent in the present data, leaving a global noise component for the fMRI analysis.

The observation that the velocity on the arterial side exhibits changes alongside respiratory variations is consistent with the findings of previous reports using transcranial Doppler ultrasonography [70,71]. This information was extracted from the BOLD lag structure, which itself presented autoregulatory response consistent with earlier work [4]. During the initial whole-brain CBV increase in response to autoregulatory vasodilation, a sole increase in the inflow should first occur to meet the volume demand. It is therefore reasonable that this effect is absent in the outlet (i.e., the venous side of the gross vasculature). This observation seems to support the model in which the BOLD lag structure is derived from an axial non-uniformity in the vessels, already present in the inflow [38]. A distinct mechanism of the lag structure was also suggested by the diminished magnitude observed in the outlet side of the vasculature, since a CBF increase should evoke larger response in the downstream [72].

It is unclear what proportion of this axial variation is systemic, i.e., originates from the autonomic loops, mediated by peripheral baro- or chemoreceptors. However, even when the neural activity is contributing to the sLFO time course as demonstrated previously [28], the resulting lag structure largely reflects the vasculature. In this work, we focused on non-neuronal mechanisms to account for the BOLD lag structure as much as possible, in the hope that it may ultimately help achieve a better understanding and provide improved modeling approaches of the fMRI signal.

### Source of BOLD low-frequency oscillation signals

Previous studies on sLFOs have reported that both Hb species fluctuate, but with varying phase differences that are selectively found in the brain [19–21]. The observed fluctuations of total Hb density have been linked to CBV changes [73–75], but interpretations for that of deoxy-Hb have rarely been provided. Only 1 series of studies by Fantini and colleagues directly addressed the possible axial variation of blood content such as oxygen saturation [76]. In

support of the conventional theory, Rayshubskiy and colleagues reported, in their human intraoperative study, that slow Hb oscillations correlated with vasomotion in the superficial arteries [21]. However, it remains unclear whether an equivalent vasomotion exists in the non-arterial vessels to fully account for the observed lag structure. Hence, it is worth considering other sources of deoxy-Hb variation.

The concept of vasomotion stems from an active diameter change in the precapillary vessels, driving local velocity fluctuations, termed "flowmotion" [77]. This flowmotion can reportedly accompany the fluctuation of local Hct that should affect deoxy-Hb density [78,79]. Another possible source for the deoxy-Hb fluctuations is a change in $SaO_2$ that ranges from 94–98% in the artery [53,80]. For example, the respiration-related BOLD signal component is supposed to be mediated by the blood $CO_2$ level and pH, which can shift the oxygen dissociation curve [35,81]. These parameters are considered to fluctuate in the blood as part of the autonomic loop, possibly driving local vasomotion, which persists after denervation [82]. As mentioned above, the phase difference between the 2 Hb species remains to be elucidated [25], but those observations do substantiate an unstable deoxy-Hb supply in brain tissues. Besides, such a signal component would have escaped detection in NVC studies using trial averaging.

A signal origin intrinsic to the flowing blood may, as suggested by Tong and colleagues, thus explain the constant phase difference among signals from different body parts [11,26]. In the literature, an axial variation of the Hct in the brain has indeed been suggested, in relation to both NVC [83–85] and sLFOs [79,86]. Furthermore, the reduction in T2* LFO amplitude in the outlet side can be explained by the high tissue deoxy-Hb density, which likely diminishes the proportional effect of intrinsic deoxy-Hb fluctuations, unless the OEF is completely coupled to this variation. Importantly, temporal dispersion alone would not fully account for the amplitude reduction, as it was only found in the venous side, despite the fact that the lag structure was tracked both up- and downstream from the global phase. These results provide good contrast with the stable $S_0$ response, reflecting its insensitivity to oxygen saturation. Although it would be too challenging to incorporate complex rheological parameters, a reconsideration of the constant deoxy-Hb assumption may help improve BOLD signal modelling.

In the hyperventilation condition, we observed a fast response to each ventilation cycle, accompanying blood pressure changes. This is consistent with reports using optimized acquisition techniques [87,88], supported by anatomical [89], as well as electrophysiological [90] studies. The premotor peak at coordinates [+56, 0, 40] was also very close to the reported activation site for volitional respiration [91]. Although the effect of respiratory movement cannot be fully excluded, the spatial pattern in S2B Fig is not that of a typical motion artifact centered on the brain surface [92]. In healthy participants, inhalation increases systemic venous return through decreased intrathoracic pressure, causing an elevation of cardiac output with some delay. In contrast, exhalation is considered to cause CBV increases through elevated cerebral venous pressure. To our knowledge, the timing order of these events has not been studied at the precision of the current data; further studies are needed to determine the source of this $S_0$ fluctuation [59]. The only available clue in our results is the spatial distribution, such as the interesting anterior-posterior segmentation (resembling the unique "$S_0$ lag structure" in Fig 9a) or the symmetrical pattern in the deep brain structures. Nonetheless, some mechanical effects of the respiratory act on the fluid dynamics likely exist, causing this spatially heterogeneous $S_0$ deflection.

## Lag structure as noise

Based on the assumption that the global signal fluctuation is the sum of all variations by NVC, its elimination by GSR has been considered to negatively bias the results [93]. However, as

noted by Aguirre and others, there are cases where GSR yields interpretable results even in the absence of global motion artifacts [41]. The rise in popularity of rs-fMRI since 2005 has led to this issue resurfacing in a different form. A variation in GSR, in which the time series extracted from a set of regions-of-interest (whole brain, white matter, and cerebrospinal fluid) are removed, has become a *de facto* standard. It is indeed computationally closer to our deperfusioning in that the regional phase difference is somehow tolerated. However, because this practice lacks a strong theoretical background [7,40], currently, the identification and elimination of bodily movements and physiological noise are more widely recommended. There are various approaches to this end, such as simultaneous physiological measurements [94], as well as data-driven methods that only use fMRI data [95]. To date, however, objective criteria for distinguishing neural activity from noise components remains an issue [96].

In the present study, the lag structure was treated as a broadly distributed, structured noise for fMRI. Indeed, it can be partly eliminated by sophisticated denoising techniques [28]. However, the specificity of lag mapping in isolating information on a purely vascular origin remains unclear. For example, measurements of velocity changes critically depend on a recursive lag-tracking method that incorporates the gradual change in LFO over regions [97]. Adaptation for changes in the waveform that may arise from different paths of the blood was demonstrated to increase the reproducibility of the lag map [28]. However, as the changes in waveform can also reflect NVC, removing the whole lag structure may lead to type II errors in the fMRI results. Hence, the favorable impacts of the deperfusioning procedure that we observed on the fMRI results are clearly insufficient to prove the advantage of this technique and require further confirmation.

Importantly, the detection of the lag structure itself largely depends on the data quality, especially in terms of head movement. When a head movement results in a synchronized deflection that exceeds the LFO amplitude, it would obscure the phase variation. However, it can be also questioned if the correlational structure of neural activity is reliably detected from such motion-contaminated data. In general, NVC should have a limited spatial extent and signal magnitude without time-locked averaging [98]. In turn, successful tracking of a lag structure may even be considered as evidence of "clean" data. The elimination of this identified lag structure can be a relatively straightforward approach to reduce structured physiological noise [93].

In conclusion, by investigating various aspects of the BOLD sLFO, we compiled supporting evidence for a component intrinsic to flowing blood that has been a focus of interest in earlier works [11]. To establish a framework by which the fMRI signal can be fully modeled, more detailed characterization of the lag structure as part of the "global noise" is needed [99].

## Supporting information

**S1 Fig. Temporal analysis of the signal components. A**, Correlation analysis of the multi echo-derived signals extracted from the three vascular regions. Effects of vascular region was observed on both $S_0$ and $T2^*$ components, but in a different manner. **B**, Phase analysis of the low-frequency component below 0.1 Hz. Phase delay relative to BOLD time course was calculated for the two $T2^*$ weighted signals. Respiratory challenges enhanced the phase difference between $S_0$ and $T2^*$. **C**. The same data as in B, but separately plotted for each region. Dissociation of the $T2^*$ and $S_0$ phase, as well as its interaction with the vascular region is evident. All these effects were small with spontaneous low-frequency fluctuation but enhanced in artificial oscillation by respiratory challenges.
(PDF)

**S2 Fig. Signal profiles in the fMRI clusters and respiration-related $S_0$ signal change. A**, Raw signal responses in the motor/premotor activation cluster for each task from Experiment 2. Time courses were resampled to a sampling interval of 0.5 s. Small but distinct responses were found in the non-BOLD or $S_0$ component but with a dominating fast component compared to the regional response, apparently independent of both the neurovascular coupling and hemoglobin fluctuations that accompany $T2^*$ changes. This response was roughly out of phase with the beat-to-beat mean arterial blood pressure shown in Fig 3, although the blood pressure response was absent during the first two cycles. Arrows indicate the fluctuation corresponding to the neurovascular coupling modelled in the SPM analysis. **B**. Group SPM to assess the distribution of the respiratory phase-related fast fluctuation of $S_0$ during hyperventilation. A separate SPM analysis was performed by modeling the fast $S_0$ fluctuation only. An anterior-posterior segmentation is evident. In addition, there are small symmetrical structures along the major veins. **C**. Additional experiment from a subset of participants (N = 7) using a different set of repetition time/flip angle to manipulate the T1-inflow effect on the $S_0$ component. The overall $S_0$ response is lower than those shown in panel **A**, but the fast response to hyperventilation is preserved.
(PDF)

## Acknowledgments

We would like to thank Editage (www.editage.jp) for English language editing.

## Author Contributions

**Conceptualization:** Toshihiko Aso.

**Data curation:** Toshihiko Aso.

**Formal analysis:** Toshihiko Aso, Shinnichi Urayama.

**Funding acquisition:** Toshihiko Aso, Toshiya Murai.

**Investigation:** Toshihiko Aso, Shinnichi Urayama.

**Methodology:** Toshihiko Aso, Shinnichi Urayama.

**Resources:** Toshiya Murai.

**Software:** Toshihiko Aso.

**Supervision:** Hidenao Fukuyama, Toshiya Murai.

**Visualization:** Toshihiko Aso.

**Writing – original draft:** Toshihiko Aso, Toshiya Murai.

**Writing – review & editing:** Shinnichi Urayama, Hidenao Fukuyama, Toshiya Murai.

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
