## [Decision Letter · Decision Letter 0]

20 Aug 2019

PONE-D-19-19654

Axial variation of deoxyhemoglobin density as a source of BOLD low-frequency time lag structure.

PLOS ONE

Dear Dr Aso,

Thank you for submitting your manuscript to PLOS ONE. After careful consideration, we feel that it has merit but does not fully meet PLOS ONE’s publication criteria as it currently stands. Therefore, we invite you to submit a revised version of the manuscript that addresses the points raised during the review process.

Some figures and references need to be revised.

We would appreciate receiving your revised manuscript by Oct 04 2019 11:59PM. To enhance the reproducibility of your results, we recommend that if applicable you deposit your laboratory protocols in protocols.io, where a protocol can be assigned its own identifier (DOI) such that it can be cited independently in the future. For instructions see: http://journals.plos.org/plosone/s/submission-guidelines#loc-laboratory-protocols

We look forward to receiving your revised manuscript.

Kind regards,

Quan Jiang, Ph,D.

Academic Editor

PLOS ONE

Journal Requirements:

1. We note that you have stated that you will provide repository information for your data at acceptance. Should your manuscript be accepted for publication, we will hold it until you provide the relevant accession numbers or DOIs necessary to access your data. If you wish to make changes to your Data Availability statement, please describe these changes in your cover letter and we will update your Data Availability statement to reflect the information you provide.

2. Please amend either the title on the online submission form (via Edit Submission) or the title in the manuscript so that they are identical.

Reviewers' comments:

Reviewer's Responses to Questions

**Comments to the Author**

1. Is the manuscript technically sound, and do the data support the conclusions?

Reviewer #1: Yes

Reviewer #2: Yes

2. Has the statistical analysis been performed appropriately and rigorously? 

Reviewer #1: Yes

Reviewer #2: Yes

3. Have the authors made all data underlying the findings in their manuscript fully available?

Reviewer #1: No

Reviewer #2: Yes

4. Is the manuscript presented in an intelligible fashion and written in standard English?

Reviewer #1: Yes

Reviewer #2: Yes

5. Review Comments to the Author

Reviewer #1: The research is original. Experiments are well designed and the conclusions are supported by the data. I appreciated the extensive use of citations to back up the introduction and discussion sections.

Minor comments:

Although all the details of the figures are given in the text and in the legends, some figures would benefit from a few additions:

Fig1: There could be a confusion in Part A between the spatial extent of vasculature and the time variations in 1-3. It would be clearer if the time arrows were bigger. I didn’t understand the usefulness of the second schematic in gray area.

Fig2: Putting a legend related to the colorlines in partB woud help.

Fig3: Part A. This was especially difficult to follow. The colorbars are missing as well as the axis names. I would remove the resampling part and add graphs about instantaneous phase differences times series and the corresponding averages. Part B. There is no colorbar to explain the different line colors.

Some references are not properly cited:

Hoge R, Atkinson J, Gill B, Crelier G & Marrett S (1999). Investigation of BOLD signal

dependence on cerebral blood flow and oxygen consumption: The …. Magnetic

Resonance in Medicine 863, 849–863. Misssing part of the title

Kennerley AJ, Berwick J, Martindale J, Johnston D, Papadakis 868 N & Mayhew JE (2005).

Concurrent fMRI and Optical Measures for the Investigation of the Hemodynamic

Response Function. 365, 354–365. No Journal title

Willie CK, Tzeng Y-C, Fisher JA & Ainslie PN (2014). Integrative regulation of human

brain blood flow: Integrative regulation of human brain blood flow. The Journal of

Physiology 592, 841–859. repetition in the title

Reviewer #2: This is an important and exciting contribution that not only acknowledges the vascular-derived lag-based structure inherent in the BOLD response but proposes a thoughtful method for combining this information with an optimized global signal regression approach that shows promise in reducing inter-individual variation while preserving task-based BOLD responses to visuomotor and respiratory tasks. The analyses and statistical approach was rigorously conducted with appropriate treatment of multiple comparisons. I did not find any methodological errors or unreasonable interpretations of the data that should slow publication of what could be a new technical standard in treatment of fMRI signal data.

6. PLOS authors have the option to publish the peer review history of their article (what does this mean?). If published, this will include your full peer review and any attached files.

Reviewer #1: No

Reviewer #2: Yes: Jeffrey S Anderson

---

## [Author Response · Author response to Decision Letter 0]

28 Aug 2019

Responses to reviewers

We thank the reviewers for the encouraging and constructive comments. In addition to the modifications below, the main text and the citation style were re-formatted to meet the journal’s standards.

Reviewer #1

>Fig1: There could be a confusion in Part A between the spatial extent of vasculature and the

> time variations in 1-3. It would be clearer if the time arrows were bigger. I didn’t understand

> the usefulness of the second schematic in gray area.

Thanks to the Reviewer, we noticed that there was much room for improvement in the Figures. As pointed out, the arrows in the top panels indicated blood flow/vasomotion (over space), while that in the bottom panel indicated time. We removed the arrows from the panels 1-3, illustrating the vessel content, which seemed redundant. The broken rectangle in the top panel was also too big (vertically) and indeed confusing for the readers to understand the three panels. The text in the gray box was changed to clearly mention the purpose of inserting this scheme. This was to explain the difference between lag mapping and perfusion MRI using contrast agent. Figure legend was also modified for clarity.

>Fig2: Putting a legend related to the colorlines in partB woud help.

>Fig3: Part A. This was especially difficult to follow. The colorbars are missing as well as 

>the axis names. I would remove the resampling part and add graphs about instantaneous phase

>differences times series and the corresponding averages. Part B. There is no colorbar to explain

>the different line colors.

We modified the figures accordingly. Part A was changed to clearly indicate that the combination of a lag map and the corresponding time courses constitute a lag structure for each individual fMRI run (shaded rectangle). An example of instantaneous phase difference timeseries (after time-locked averaging) was inserted. Colorbar was added to the Fig 3B.

Citations are fixed, thank you.

---

## [Decision Letter · Decision Letter 1]

9 Sep 2019

[EXSCINDED]

Axial variation of deoxyhemoglobin density as a source of the low-frequency time lag structure in blood oxygenation level-dependent signals

PONE-D-19-19654R1

Dear Dr. Aso,

We are pleased to inform you that your manuscript has been judged scientifically suitable for publication and will be formally accepted for publication once it complies with all outstanding technical requirements.

With kind regards,

Quan Jiang, Ph,D.

Academic Editor

PLOS ONE

Additional Editor Comments (optional):

Reviewers' comments:

Reviewer's Responses to Questions

**Comments to the Author**

1. If the authors have adequately addressed your comments raised in a previous round of review and you feel that this manuscript is now acceptable for publication, you may indicate that here to bypass the “Comments to the Author” section, enter your conflict of interest statement in the “Confidential to Editor” section, and submit your "Accept" recommendation.

Reviewer #1: All comments have been addressed

Reviewer #2: All comments have been addressed

2. Is the manuscript technically sound, and do the data support the conclusions?

Reviewer #1: Yes

Reviewer #2: Yes

3. Has the statistical analysis been performed appropriately and rigorously? 

Reviewer #1: Yes

Reviewer #2: Yes

4. Have the authors made all data underlying the findings in their manuscript fully available?

Reviewer #1: Yes

Reviewer #2: Yes

5. Is the manuscript presented in an intelligible fashion and written in standard English?

Reviewer #1: Yes

Reviewer #2: Yes

6. Review Comments to the Author

Reviewer #1: (No Response)

Reviewer #2: Authors have appropriately addressed reviewer concerns. I have no further concerns that require addressing.

7. PLOS authors have the option to publish the peer review history of their article (what does this mean?). If published, this will include your full peer review and any attached files.

Reviewer #1: No

Reviewer #2: No

---

## [Editor Report · Acceptance letter]

11 Sep 2019

PONE-D-19-19654R1 

Axial variation of deoxyhemoglobin density as a source of the low-frequency time lag structure in blood oxygenation level-dependent signals 

Dear Dr. Aso:

I am pleased to inform you that your manuscript has been deemed suitable for publication in PLOS ONE. Congratulations! Your manuscript is now with our production department. 

With kind regards,

on behalf of

Dr. Quan Jiang 

Academic Editor

PLOS ONE